# A Traceable DNA-Replicon Derived Vector to Speed Up Gene Editing in Potato: Interrupting Genes Related to Undesirable Postharvest Tuber Traits as an Example

**DOI:** 10.3390/plants10091882

**Published:** 2021-09-10

**Authors:** Giovana Acha, Ricardo Vergara, Marisol Muñoz, Roxana Mora, Carlos Aguirre, Manuel Muñoz, Julio Kalazich, Humberto Prieto

**Affiliations:** 1Programa de Doctorado en Biotecnología, Universidad de Santiago, Santiago 9170020, Chile; giovana.acha@usach.cl; 2Laboratorio de Biotecnología, Instituto de Investigaciones Agropecuarias-La Platina, Santiago 8831314, Chile; mmunoz@inia.cl (M.M.); rmora@inia.cl (R.M.); caguirre.d@gmail.com (C.A.); 3Instituto de Investigaciones Agropecuarias-Remehue, Osorno 5290000, Chile; manuel.munozd@inia.cl; 4Carrera de Agronomía, Campus Osorno, Universidad de Los Lagos, Osorno 5290000, Chile; julio.kalazich@ulagos.cl

**Keywords:** potato, CRISPR/Cas9, genome editing, geminivirus replicons, agrobacterium-mediated transformation, green fluorescent protein

## Abstract

In potato (*Solanum tuberosum* L.), protoplast techniques are limited to a few genotypes; thus, the use of regular regeneration procedures of multicellular explants causes us to face complexities associated to CRISPR/Cas9 gene editing efficiency and final identification of individuals. Geminivirus-based replicons contained in T-DNAs could provide an improvement to these procedures considering their cargo capability. We built a *Bean yellow dwarf virus*-derived replicon vector, pGEF-U, that expresses all the editing reagents under a multi-guide RNA condition, and the *Green Fluorescent Protein* (*GFP*) marker gene. *Agrobacterium*-mediated gene transfer experiments were carried out on ‘Yagana-INIA’, a relevant local variety with no previous regeneration protocol. Assays showed that pGEF-U had *GFP* transient expression for up to 10 days post-infiltration when leaf explants were used. A dedicated potato genome analysis tool allowed for the design of guide RNA pairs to induce double cuts of genes associated to enzymatic browning (*StPPO1* and *2*) and to cold-induced sweetening (*StvacINV1* and *StBAM1*). Monitoring GFP at 7 days post-infiltration, explants led to vector validation as well as to selection for regeneration (34.3% of starting explants). Plant sets were evaluated for the targeted deletion, showing individuals edited for *StPPO1* and *StBAM1* genes (1 and 4 lines, respectively), although with a transgenic condition. While no targeted deletion was seen in *StvacINV1* and *StPPO2* plant sets, stable GFP-expressing calli were chosen for analysis; we observed different repair alternatives, ranging from the expected loss of large gene fragments to those showing punctual insertions/deletions at both cut sites or incomplete repairs along the target region. Results validate pGEF-U for gene editing coupled to regular regeneration protocols, and both targeted deletion and single site editings encourage further characterization of the set of plants already generated.

## 1. Introduction

Genome editing strategies in plants based on the Clustered Regularly Interspaced Short Palindromic Repeats and (CRISPR)-Associated Protein 9 System (CRISPR/Cas9) rely on three fundamental aspects: (a) adequate delivery systems for the editing reagents (i.e., guide RNA (gRNA) and Cas9 nuclease), (b) tissue culture procedures that allow for regeneration of the edited cells, and (c) genome knowledge of the cultivar to be edited, avoiding off-target loci effect by the reagents. 

In potato (*Solanum tuberosum* L.), different approaches have been used to achieve successful CRISPR/Cas9-based genome editing. Conventional T-DNA vectors were first used on leaf explants to generate transgenic T1 lines of ‘DM’ (a ‘doubled monoploid’ potato clone of the group Phureja DM1-3 516 R44, hereafter referred to as ‘DM’) genotype, to express the editing reagents allowing for mutated versions of the *AUXIN/INDOLE-3-ACETIC ACID PROTEIN* gene (*StIAA2*) [1]. More recently, this type of vectors allowed for CRISPR/Cas9 targeted mutations in the promoter region of *StFLORE*, a long non-coding RNA, and showed the regulator role of this molecule in flowering and drought tolerance [2]. These vectors led to the generation of transgenic plants expressing the editing reagents; however, the transgenic condition constitutes an additional bottleneck in the process considering that the species is highly heterozygous and most of their cultivars show gametophytic self-incompatibility, making transgene segregation difficult by selfing or out-crossing techniques. Genome editing by CRISPR/Cas9 has also been achieved by coupling protoplast techniques to delivery of pre-assembled editing reagents (i.e., as ribonucleoproteins) in ‘Kuras’ [3], conducting the complete knockout of the four alleles of the *GRANULE BOUND STARCH SYNTHASE* gene (*StGBSS*). In ‘Désirée’, this approach was used to induce mutations in the *STARCH-BRANCHING ENZYME 1* and *2* [4] or in the *POLYPHENOL OXIDASE 2* [5] genes, leading to the generation of plants producing tubers with starch essentially lacking branching or showing reduced enzymatic browning, respectively. However, success in protoplast protocols is far from routine in the current plant tissue culture state of the art, and regular gene transfer procedures, based mostly on regeneration protocols from multicellular explants such as leaves, internodes, and somatic embryos, represent a conditioning factor for gene editing which is under continuous improvement [6]. In this regard, new approaches for expressing the editing reagents coupled with regular gene transfer procedures using multicellular explants could contribute to massive use of the technique. 

At the same time that recent studies have shown that viral vectors may be useful expressing the editing reagents in regular explants used in *Agrobacterium*-mediated gene transfer [7], important advances in the delivery of these tools have also been achieved using T-DNAs containing autonomously replicating geminivirus vectors (LSL vectors) [8]. Characterized by an important cargo capability [9], these vectors include all the essential viral elements from the virus replication machinery retained in the T-DNA, allowing virus replication by the rolling circle (RC) mechanism to be emulated and, thus, enabling the transcriptional activation of the included expression cassettes. These elements include the long and short intergenic regions (LIR and SIR, respectively), required as regulator (SIR) and structural (LIR) sequences for adequate RC function [10,11]. Consequently, the virus replication initiator protein (Rep/RepA) is also required [12] and is regularly supplied by the viral replicon itself or must be externally provided [11,13]. During *Agrobacterium*-mediated gene transfer experiments, the LSL components included in the T-DNA are activated by Rep/RepA acting on the LIR components of the vector, leading to the replicative release of the recombinant DNA included between them [13]. For gene editing vectors, this includes the information for Cas9 and the gRNAs [9]. The released DNA then replicates episomally in the nucleus, leading to the efficient expression of that encoded information [14]. Thus, the gene editing power relies on the expression of the editing reagents—i.e., Cas9 and gRNAs—and on the time by which these elements are generated in the cell. The use of geminivirus T-DNA vectors containing the cis-acting LSL components allowed for gene editing of potato leaf explants of ‘Désirée’ and in a diploid self-incompatible breeding line derived from ‘DM’. These experiments demonstrated their effectiveness in generating inheritable mutations of the *ACETOLACTATE SYNTHASE 1* gene (*StALS1*) in either the absence [15] or the presence of a repair template to generate the mutations [16]. More recently, general procedures for potato gene editing by assembling geminivirus-derived replicons containing the editing reagents and eventually repair templates, have been detailed [9].

While these results have shown the feasibility of carrying out gene editing of potato by geminivirus T-DNA vectors, this experimentation has also revealed weak points of the strategy which are related mainly to the efficiency for the gene targeting and in the traceability of the mutation from the starting explant (calli) to the final edited individual (primary edited individual and stable edited segregant) [15,16].

The high self-replicative rate of the LSL sequences incorporated in the T-DNA vectors could represent important advantages in potato genome editing; in addition to a stable integration of the elements present in the T-DNA, the LSL sequences could efficiently and transiently express these reagents in events even with no exogenous DNA insertion into the plant genome [8,15,17]. This latter is expected to take place in highly efficient transient expression events during the first steps of the explant’s regeneration. Thus, monitoring of the self-replicative and cassette expression activities in these vectors could improve the gene editing process, at least as a preliminary selection step undergone under the transient phase of a gene transfer experiment. In this regard, improvement in selection of edited cells has been achieved by fluorescence cell sorting of edited protoplasts isolated from agroinfiltrated *Nicotiana benthamiana* leaves using viral editing vectors which also include the *Green Fluorescent Protein* (*GFP*) gene [18].

In this work, we describe the design and evaluation of pGEF-U, an LSL-based T-DNA vector aimed at enabling genome editing at the time when a maker, the *GFP* gene, allowed for vector-cell interaction monitoring during the process. To evaluate this system, we targeted potato genes associated with unwanted postharvest events of tubers related to enzymatic and non-enzymatic browning. The current literature associates the activity of POLYPHENOL OXIDASE (PPO) enzymes StPPO1, StPPO2, StPPO3, and StPPO4 with enzymatic browning [5,19]. Regarding non-enzymatic browning, there is evidence that it is associated with the cold-induced sweetening process (CIS), a phenomenon to which the activity of the VACUOLAR INVERTASE 1 (StvacINV1) enzyme and the beta- and alpha-AMYLASES StBAM1, StBAM9, and StAmy23 enzymes are associated [20,21]. Particular pGEF-U versions were built and evaluated for both editing and traceability capabilities in ‘Yagana-INIA’, a domestic elite variety in the country for which this is the first report on regeneration and gene edition.

## 2. Results

### 2.1. Functionality of the Universal Fluorescent Editor Geminivirus-Based Plasmid (pGEF-U) in Potato

The design of a traceable LSL-type vector, pGEF-U, considered the incorporation of the LSL elements for geminivirus RC multiplication, the expression cassettes for the CRISPR/Cas9 editing reagents (i.e., Cas9 nuclease and gRNAs), and the expression cassette for the *GFP* gene (Figure 1a). As shown in Figure 1b, pGEF-U has a site for cloning modules containing up to four gRNAs between the *Bsa*I recognition sites. The functionality of this new LSL-type vector was assayed by *Agrobacterium*-mediated transformation of ‘Yagana-INIA’ leaf and internode explants. As shown in Figure 2a, transient expression of pGEF-U allowed for the expression of the GFP marker at 7 days post-infiltration (dpi). An average of 58.2% of leaf explants was infected, whereas the use of internodes showed only 4.4% of explants expressing GFP (Appendix A). Therefore, from now on, we decided to use only leaf explants for validating gene editors (i.e., gene editing versions of pGEF-U, pGEF-X, where X is the target gene).

### 2.2. Guide RNAs Design and Evaluation

In addition to the inclusion of the *GFP* marker gene, pGEF-U keeps the multi-gRNA scaffold module (Figure 1b) allowing for a gene editing strategy aimed at the inactivation of target genes by removal of gene fragments. Based on the Potato Reference Genome from the homozygous ‘DM’, a dedicated genome processing tool was built (see CRISPR Search; www.fruit-tree-genomics.com/biotools, accessed on 20 March 2018) and used for gRNA design. To inactivate the target genes, pairs of gRNAs were selected from exonic regions of both the CIS-related genes *StvacINV1* and *StBAM1* and the *PPO* target genes *StPPO1* and *StPPO2* and cloned into pGEF-U (Appendix A). The information for the code locations of each gene is shown in Appendix A. The selected gRNA pairs are summarized and characterized in Table 1, which also describes their nucleotide sequences and positions within genes, as well as the expected size of deletions by the double cuts.

The identity of the modeled gRNAs from the ‘DM’ genome-based tool and the actual targets in the ‘Yagana-INIA’ background were experimentally determined by sequencing their corresponding target genomic regions (Appendix A). The names and positions of the primers used for each case are shown in the corresponding schemes of the genes related to both CIS and enzymatic browning (Appendix A). Primer sequences for isolation of the surrounding gRNA sites are listed in Appendix A (primers CE).

Four pGEF-X gene editors, two CIS-associated (pGEF-StvacINV1 and pGEF-StBAM1) and two *PPO*-associated (pGEF-StPPO1 and pGEF-StPPO2), were assembled. The primers allowing for cloning of the gRNA1-gRNA2 modules for each gene cloned into pGEF-U are detailed in Appendix A. The editing capability of the gRNAs contained in each pGEF-X vector—i.e., deletions of genomic regions of each gene as depicted in Appendix A—was assayed by transient gene transfer experiments in ‘Yagana-INIA’ leaf explants and tracked at 7 dpi. Three to five explants showing the highest GFP emission (Figure 2b) were subjected to PCR amplification of their genomic DNAs using CE-primer pairs (Appendix A).

The generation of smaller bands, in comparison to each original gene version, confirmed the target gene edition (Appendix A). In addition, the sequence of these amplicons validated the target gene deletions as shown in Figure 3a–d.

### 2.3. ‘Yagana-INIA’ Regeneration and Gene Editing

The implementation of a regeneration procedure for ‘Yagana-INIA’ was based on previous descriptions in ‘Désirée’ [23]. After adjustments, we observed that the ‘Yagana-INIA’ explants responded to the respective culture media with a progressive delay as development stages advanced. In addition, inclusion of *Agrobacterium*-mediated gene transfer using these explants required approximately seven more weeks than ‘Désirée’ (Figure 4). Selected explants, judged as being able to proceed with the regeneration procedure (Figure 4a), began callusing after 5 weeks of culturing (Figure 4b), whereas that initial budding took place after 13 weeks (Figure 4c); finally, plantlets were individualized after week 24 (Figure 4d).

Gene transfer experiments using *Agrobacterium*-pGEF-X clones were then incorporated into this protocol and the process was monitored by GFP emission. Explants with the highest GFP fluorescence at 7 dpi were selected to proceed to plantlet regeneration in PSM400cc (Table 2). After three independent experiments, the positivity of the infection of ‘Yagana-INIA’ leaves based on GFP emission ranged from 19.7% to 78.3% of the total explants; however, the percentage of explants allowed to proceed and regenerate in PSM400cc ranged from 8.2% to 67.2%. This is explained because only explants with the highest extent of expression were transferred to the PSM400cc medium. After this selection, the GFP fluorescence emission was no longer detected in most of the explants, and they proceeded to callusing and from these tissues to regenerated shoots. Table 2 summarizes the number of starting explants transformed per pGEF-X gene editor, explants that yielded calli, and, finally, the number of individualized shoots. In general terms, explants judged adequate to proceed for gene transfer reached 1040, from which 580 (55.8%) presented some extent of *GFP* expression, allowing for a final selection of 357 (34.3%) highly expressing explants proper for whole plant regeneration. The regenerative capacity of ‘Yagana-INIA’ presented values from 0.1 to 6.1 shoots per callus, showing that this variety presents a wide range of regenerative responses in the PSM medium. The total number of individualized plantlets isolated 40 weeks after initial selection from three experiments per vector was 124 for pGEF-StvacINV1, 128 for pGEF-StBAM1, 141 for pGEF-StPPO1, and 170 for pGEF-StPPO2 (Table 2).

The targeted deletion editing of genes was used as a secondary and fast screening of the produced materials suitable for molecular analysis. Three hundred and eighty-five out of these 563 total plants were sampled for DNA extraction and gene editing verification by PCR (54 for pGEF-StvacINV1, 85 for pGEF-StBAM1, 101 for pGEF-StPPO1, and 145 for pGEF-StPPO2) (Figure 5a). While individuals regenerated from experiments with the gene editors pGEF-StvacINV1 and pGEF-StPPO2 did not give the expected edition (as judged by double cut) of their target genes, five individuals derived from the edition with pGEF-StBAM1 (plant lines #180, #375, #392, and #481) and pGEF-StPPO1 (plant line #464) editors were recognized (Figure 5b). Repair options and identities of these gene editions were further analyzed by sequencing. In the case of *StBAM1* edited individuals (Figure 5c), lines #375 and #392 showed the expected edition with no additional repair options. In the case of lines #481 and #180, large fragment losses were accompanied by small extra deletions (five nucleotides) at the gRNA1-recognition site. For the *StPPO1* edited line (#464), characterization PCRs evidenced two bands with the putative targeted deletion editing (Figure 5a); cloning and sequencing revealed that the smaller corresponded to the expected theoretical size with two repair events (Figure 5d): one with the expected edition (sequenced colony #2, “C02”) and another with small extra deletions at both ends (C08). Characterization PCRs in line #464 also yielded a wild type-like gene size band (Figure 5a), for which we observed punctual deletion in the gRNA1-cutting site and a punctual insertion in the gRNA2-cutting site (Appendix A, C04), or additional small deletions at the cleavage sites of both gRNAs (Appendix A, C09). Except for the selection step in the regeneration pipeline, these five edited lines did not present GFP emission throughout the process. However, additional characterization analyses of these five lines established a *GFP* transgenic status (Appendix A).

### 2.4. Gene Editing in Callus Lines

As mentioned, GFP monitoring throughout regeneration showed that most of the explants that initially expressed GFP at 7 dpi tended to decline fluorescence emission when the callusing stage was achieved. Nevertheless, in the case of editors pGEF-StvacINV1 and pGEF-StPPO2 (Table 3), calli with stable GFP emission were observed (9 callus lines, approximately 39 weeks after transformation) (Figure 6a) and analyzed for their editing status. Four out of six callus lines derived from pGEF-StvacINV1 experiments and one out of three callus lines from pGEF-StPPO2 experiments presented shortened bands as the expected targeted deletion edition (Figure 6b); also, alternative editing and repair options, evidencing extensive deletions and/or insertions with different degrees of homology, were seen (Figure 6c,d; Appendix A). These transgenic calli represented 12.5% and 5.3% of the total explants kept in PSM400cc medium for pGEF-StvacINV1 and pGEF-StPPO2, respectively. In these callus lines, the transgenic condition was confirmed by PCR detection of both the *GFP* transgene (Appendix A) and the released and recircularized geminivirus (Appendix A).

The extent of the mismatch between the generated gRNAs and the potato genome was also computed for the prediction of eventual off-targets. Guide-RNAs with up to three or fewer mismatches and targeting on other exonic regions in the potato genome were further analyzed (summarized in Appendix A). According to these results, only the gRNA2-StPPO2C molecule showed that, with the inclusion of two potential mismatches, and this gRNA aligned a predicted zone in the *StPPO4* gene (Appendix A). This eventual off-target activity for the gRNA2-StPPO2C molecule was checked by PCR and sequencing of the predicted off-targeted region in the callus line 12 derived from the pGEF-StPPO2 editing process, confirming the theoretical prediction by the Potato CRISPR Search Tool (Appendix A).

## 3. Discussion

Formerly used as biotechnological tools for heterologous protein expression [24], LSL vector systems have, as one of their main properties, an important cargo capability. Pioneer works in CRISPR/Cas9 gene editing using geminivirus-based vectors had included the expression of both the gRNAs and the repairing DNA templates [8,14,16]. In pGEF-U, we included all the expression cassettes for Cas9, multi-gRNA scaffold, and the reporter *GFP*. In addition, geminivirus-based replicons offer the advantage of by-pass deleterious effects from their “full virus” vector versions as they avoid eventual restrictions associated with the host range [25,26].

The selection of individuals derived from genome editing procedures is a key step in the technology and represents a limiting factor for routine CRISPR/Cas in plants. In this regard, the expression of marker genes during some stage of the experimentation has been extensively studied. Chen et al. [27] used *Beta-glucuronidase* (GUS) gene expression to track the editing of the *PHYTOENE DESATURASE* (*PDS*) gene by Cas9 and a single gRNA in tobacco explants. Regular T-DNA-based vectors and *Agrobacterium*-mediated transformation of leaf explants showed that the highest GUS activity was at 3 dpi. In this way, buds recovered at this time led to the generation of stable albino tobacco mutants with no transgene insertion. Screening of CRISPR-mediated mutants was achieved using high-throughput DNA sequencing combined with high-resolution melt analyses of PCR products. Recently, Fluorescence Activated Cell Sorting of protoplasts expressing GFP tagged CRISPR/Cas9 has been used for the enrichment of edited cell populations in *Nicotiana benthamiana* [18]. A more particular example regarding this challenging situation came up almost in parallel; Veillet et al. [28] carried out base editing by CRISPR/Cas9 in potato and tomato by applying a sequential double selection monitored system, also using regular T-DNA vectors and *Agrobacterium*-mediated gene transfer. After agroinfection, a first step involved the transient selection of the explants during two weeks in kanamycin. Afterward, the authors took advantage of targeting the *ACETOLACTATE SYNTHASE* (*ALS*) gene, for which mutants could grow in medium supplemented with chlorsulfuron. While this sequential selection led to 10% of the mutants being transgene-free, this case differs from most of the target genes for which there is no choice for the selection.

Modified T-DNA vectors have also started to be proposed in this technical railway. By an only right-border type vector, Bánfalvi et al. [29] delivered the editing reagents addressing the *PDS* gene in potato and showed that transient editing reagent expression at short 3 dpi and in a system with positive/negative selection based on kanamycin could generate transgene-free *pds* mutants with a minimum frequency of 2–10%. Nevertheless, longer selection times of agroinoculated explants showed no transgenic-free mutants. According to these findings, the use of the *Neomycin phosphotransferase* II (*NPTII*) gene (or maybe another resistance gene) in pGEF-U could improve this vector by adding the possibility that applying an antibiotic selection pulse (as in the “early selection strategy”) at the same time that GFP expression is monitored. In the current version, tracking GFP transient emission from pGEF-X vectors during the first week of experimentation was a useful tool that allowed for the management of routine experiments by reducing approximately one-third of the starting processed explants, saving workspace, time, and resources. These factors are relevant if we consider the use of regular regeneration protocols in which long-term experiments are needed before an outcome. In this line, regeneration and transformation deduced for ‘Yagana-INIA’ showed that this genotype was more recalcitrant than ‘Désirée’ (i.e., a delay of seven weeks until individualized shoots; Figure 4). Indeed, this low regenerative capability was deepened when *Agrobacterium*-mediated gene transfer procedures were included, requiring in some cases up to 40 weeks for the generation of a total of 563 putatively edited plantlets.

Our rationale in pGEF-U was to incorporate *GFP* expression as a tracking tool that could report the dynamics of events going on since agroinfection to plantlet generation, including transient expression (i.e., 3 to 12 dpi) and stable expression periods (i.e., over 20 dpi). In this regard, the use of *GFP* allowed for the adequate identification of these dynamic situations in potato without compromising the processed explants. Conversely, other candidate markers evaluated in parallel, such as the transcription factor *StMYBA1*, did not accomplish these identification requirements, probably due to changes in colored compound distribution and accumulation during the development of the explants. From 563 individuals selected and regenerated, 385 were analyzed and among them we found that 23 turned out to be transgenic for the *GFP* gene by PCR, including the five edited lines, though no GFP emission was detected in these individuals as judged by epifluorescence microscopy. A possible explanation for this could be the transcriptional and/or post-transcriptional gene silencing mechanism generated during callus development and/or subsequent plantlet regeneration. Evidence for the existence of this suppression mechanism for *GFP* gene expression has already been reported in potatoes [30]. While this condition for the use of GFP to track the final stages of regeneration schedules must be considered, these facts reinforce our idea about future improvements in pGEF-U vectors using additional selection markers to boost the transient selection approach.

Despite constant increases, there are still limits to the availability of plant genome analysis tools dedicated to CRISPR/Cas technology. In that way, gRNA design is the starting point of any research initiative. For this reason, we designed a processing pipeline for CRISPR/Cas9 feasibility specifically in potato. Although in our pipeline the design of single gRNAs is derivable, the tool has been designed to support the use of gRNA molecules as functional pairs, according to the gene inactivation approach. In comparison to our application, most of the available gRNA design tools focus on the targeted indel-based mutations by use of independent gRNAs [31,32]. In addition, gene inactivations by the removal of gene segments could represent a more efficient approach in terms of effective gene inactivation compared to the use of single gRNA [33]. The double cut condition was also taken as part of the primary screening carried out in the work. It is necessary to indicate that the generated gRNA pairs are also assessed for possible off-targets in the ‘DM’ reference genome; this characteristic could bring a complete assessment procedure for the finally identified and selected plant prototypes.

These tools were assayed on ‘Yagana-INIA’, a commercially relevant variety used for both fresh consumption and agroindustry. As a parental line in domestic breeding programs, it has been used as progenitor to obtain several widespread potato varieties as Patagonia-INIA, Pukará-INIA, and Karú-INIA [34]. This genotype contributes with traits such as higher yield and tuberizing rates, therefore is a valuable resource for breeding because its agronomic background to obtain new varieties, additionally to its importance as commercial cultivar. In this way, the opportunities to develop new market possibilities and deployment of new commercial cultivars with high agronomic value and better cooking performances are enforced. We focused on the inactivation of genes related to two unwanted phenomena that take place during the postharvest storage of potato tubers. Enzymatic browning has been associated mainly with PPO activity generated by gene expressions of *StPPO1* to *StPPO4* [19]. Consequently, the use of artificial microRNAs allowed for the simultaneous silencing of these isoforms, leading to an important reduction in tubers’ browning [19]. Recently, CRISPR/Cas9 gene editing of the *StPPO*2 isoform was achieved by delivery of the preassembled editing reagents to ‘Désirée’ protoplasts and confirmed this isoform as being of main relevance in the process, producing tubers with 73% less enzymatic browning [5]. In our work, pGEF-StPPO1 and pGEF-StPPO2 targeted the two most likely relevant *PPO* isoforms involved in the process, *StPPO1* and *StPPO2* [19] and the results showed the efficacy of the corresponding pGEF-X vectors in editing those genes. In this work, we identified one edited line for *StPPO1* that will allow functional evaluations to begin once tuber generation is achieved, revealing the power of this procedure for gene editing in potato. The same is expected for those shoots that could regenerate from *StPPO2* edited callus; according to our analyses, callus line 12 was predicted by our “Potato CRISPR Search Tool” to present an off-target at *StPPO4* gene. In this case, two mismatches are located outside the “seed region” of one of the used gRNAs (Appendix A). Since number and position of these mismatches have been described as critical factors for this type of evaluations [35], functional evaluation of both *StPPO2* (on-target) and *StPPO4* (off-target) edited materials could be relevant in future works.

On the other hand, the accumulation of reducing sugars (fructose and glucose) in potato tubers during CIS is associated with metabolic disorders in processes such as starch synthesis and degradation, glycolysis, hexogenesis, and mitochondrial respiration [36]. Among the enzymes involved in these disorders highlight vacuolar invertases [37] and β-amylases [21]. Genetic transformation of potato ‘E3′ with RNAi constructs targeting candidate *BETA-AMYLASE* genes showed that silencing of the *StBAM1* could lead to decreased BETA-AMYLASE activity in cold-stored tubers [21]. Similarly, the use of transcription activator-like effector nucleases (TALENs) to knockout *vacINV* in ‘Ranger Russet’ produced undetectable levels of reducing sugars in the selected individuals [20]. The use of pGEF-StvacINV1 and pGEF-StBAM1 in our work showed that they were able to edit their targets, and the edited materials for the *StBAM1* (four plant lines) and the *StvacINV1* (four callus lines) genes offer the possibility of carrying out further functional phenotyping of tubers in the near future.

While the primary selection of targeted deletion edited individuals for two (*StBAM1* and *StPPO1*) of the four initially selected genes was successfully achieved, some relevant considerations were raised from our analyses in both edited plants and calli showing stable GFP expression. Characterization of line #464 (*StPPO1*) and callus line 12 (*StPPO2*), showed that bands corresponding to the non-double cut gene version, based on their amplification size by PCR, finally corresponded to gene versions containing indels at each of the cutting sites by gRNA1 and gRNA2 (Appendix A). Based on these findings, the already generated materials could require further analyses addressing the detection of indels due to each gRNA from the vector. While the use of pGEF-U is validated in this study for gene editing in potato and, eventually, other plant species, these results encourage us to further analyze these populations under broader criteria, including mutations caused by each single gRNA in these apparently ‘native’ gene versions. In this line, several fast techniques, for instance, IDAA (indel detection by amplicon analysis) [38], show up as feasible procedures to be carried out for our short-term future steps in this study. Interestingly, due to the mechanism involving the pGEF-U expression, the possibility of finding non-transgenic edited individuals in these populations is attractive.

## 4. Materials and Methods

### 4.1. Plant Material

Potato cv. Yagana-INIA is an elite cultivar from the Potato Breeding Program at Remehue Experimental Station, INIA-Chile. ‘Désirée’ seeds were obtained from the Germplasm Bank in La Platina Station, INIA-Chile. Materials were propagated in vitro in glass jars (40 × 120 mm) containing 25 mL of Potato Propagation Medium (PPM) (Appendix A). Nodal stem pieces of 0.5–1.0 cm were used for propagation every 4–6 weeks. Potato plantlets were cultured in a growth chamber with a photoperiod of 16 h light/8 h dark (white fluorescent light) at 25 °C.

### 4.2. Construction of the Geminivirus Editor Fluorescent Universal Plasmid (pGEF-U) for Potato Gene Editing

The pGEF-U vector includes expression cassettes for *GFP*, Cas9, and up to four independent gRNAs. The GFP expression cassette was first built by CaMV 35S promoter PCR amplification using the primers attB1-CaMV35Sx2/attB2-CaMV35Sx2 (Appendix A) and the pGWB402 vector as a template (Addgene plasmid #74796). PCR reactions consisted of 5 µL attB1-CaMV35Sx2/attB2-CaMV35Sx2 primers mix (10 μM), 10 µL 5× SuperFi Buffer (Thermo Fisher Scientific, Waltham, MA, USA), 1 µL dNTPs (10 mM), 1 µL pGWB402 vector (10 ng/µL; Addgene Plasmid #74796), 0.5 µL Platinum SuperFi DNA Polymerase (Thermo Fisher Scientific), and 32.5 µL nuclease-free water, resulting in a final volume of 50 µL. PCR conditions were: initial denaturation 98 °C (30 s), 40 cycles (98 °C (10 s), 58 °C (10 s), 72 °C (45 s)) and final extension of 72 °C (10 min). The amplicon was recombined into the pDONR207 donor vector (Thermo Fisher Scientific) following the manufacturer’s indications and the recombination mix used in the transformation of One Shot^®^ TOP10 (Thermo Fisher Scientific) chemically competent *E. coli* cells. Recombined pDONR207-CaMV35Sx2 plasmid was used in a new recombination reaction with the pGWB504 (Addgene plasmid #74846), and this recombination mix was used in One Shot^®^ TOP10 transformation. Recombined pGWB504-CaMV35Sx2 expression clone was confirmed by sequencing (Macrogen Inc., Seoul, Korea). From this latter construct, the expression cassette consisting of CaMV35S-GFP-tNOS was amplified and isolated using the GWcst-Fw/GWcst-Rv primer pair (Appendix A). The amplification conditions were as above. PCR reactions consisted of 5 µL GWcst-Fw/GWcst-Rv primers mix (10 μM), 10 µL 5× SuperFi Buffer, 1 µL dNTPs (10 mM), 1 µL pGWB504-CaMV35Sx2 vector (10 ng/µL), 0.5 µL Platinum SuperFi DNA Polymerase, and 32.5 µL nuclease-free water, resulting in a final volume of 50 µL. PCR conditions were: initial denaturation 98 °C (30 s), 40 cycles (98 °C (10 s), 58 °C (10 s), 72 °C (45 s)) and final extension of 72 °C (10 min). The PCR product was electrophoresed and purified from ethidium bromide-stained 1.2% agarose gels with the ZymoClean Gel Recovery kit (Zymo Research, Irvine, CA, USA), eluted with 6 µL nuclease-free water and used for cloning into a previously built linearized LSL-type vector, pGMV-U (Addgene #112797; Appendix A) [22]. pGMV-U was linearized with the *Asc*I (*Sgs*I) FastDigest (Thermo Fisher Scientific) and dephosphorylated using the Quick Dephosphorylation Kit (New England Biolabs, Ipswich, MA, USA). Ligation of the linearized vector and the *GFP* expression cassette was carried out using T4 DNA Ligase (New England Biolabs). All these steps were carried out according to the respective manufacturers’ instructions. The ligation mix was used to transform One Shot^®^ Top 10 *E. coli* chemically competent cells and selected on 100 mg/L kanamycin LB agar plates, following the manufacturer’s instructions. Clones were checked by colony PCR, using Kapa Taq DNA Polymerase (Kapa Biosystems, Wilmington, MA, USA) according to the manufacturer’s specifications. To verify the cassette insertion within the *Asc*I restriction site, we used the primers pair *Asc*I-Upstream/*Asc*I-Downstream described in Appendix A. The determination of the direction of the insertion was verified by sequencing (Macrogen Inc.).

### 4.3. Gene Transfer Experiments

#### 4.3.1. Agrobacterium Preparation

*A. tumefaciens* strain EHA105 was electroporated using 50–100 ng of each plasmid following the procedures described by M^c^Cormac et al. [39] using a Gene Pulser II System (Bio-Rad, Hercules, CA, USA) set at 400 Ω, 1.25 kV, and 25 µF. *Agrobacterium tumefaciens* clones transformed with each expression vector were cultured in 30 mL of liquid LB medium with kanamycin (100 mg /L) for 24 h at 28 °C and 180 rpm. Bacteria were centrifuged at 4300 rpm for 10 min at 25 °C and the supernatant discarded. Bacteria were resuspended in 30 mL of PCM Liquid Medium (PLM, Appendix A) supplemented with acetosyringone (AS) (200 µM) and adjusted to an OD_600_ between 0.5–0.7. The culture was kept for 0.5 to 4 h at 28 °C and 180 rpm until use.

#### 4.3.2. Explants Pre-Culture

Internodes and leaves from 4- to 6-week-old in vitro plants were used. The day before transformation, explants were cut in pieces of approximately 0.5–1 cm. A total of 50–100 pieces of each explant were pre-cultured in a Falcon tube (50 mL) containing 25 mL of High Hormone Preculture medium (HH; Appendix A) [40] and kept in darkness until infection.

#### 4.3.3. Explants’ Infection, Co-Culture, and Callus Induction

Infections were as described in Craze et al. [23] with modifications. Pre-cultured explants were incubated in the activated Agrobacterium solution (30 mL PLM plus AS 200 µM) for 30 min with gentle agitation (50–70 rpm). Explants were decanted and the supernatant removed. Carefully, the explants were placed on solid Co-culture Potato Callusing Medium (Co-PCM; Appendix A) and kept for 2–3 d at 25 °C in a 16 h/8 h (light/darkness) photoperiod. Explants were washed twice with distilled water (30 mL each) and incubated twice with distilled water (or PLM) with cefotaxime and carbenicillin (400 mg/L each) for 30 min. Explants were dried on sterile filter paper and transferred Petri dishes containing Potato Callusing Medium with carbenicillin 400 mg/L and cefotaxime 400 mg/L (PCM400cc; Appendix A) for 5–7 d at 25 °C in the same photoperiod.

### 4.4. Plant Regeneration

For whole plant generation, explants were transferred from PCM400cc and cultured in Shooting Induction Medium PSM400cc (Potato Shooting Medium with carbenicillin 400 mg/L and cefotaxime 400 mg/L) (Appendix A) for two weeks. Additional cultures were carried out in PSM and refreshed every three weeks; to avoid bacteria traces in these cultures, staggered decreasing in antibiotic concentrations up to 200 mg/L (both carbenicillin and cefotaxime; PSM200cc) was applied. Once shootings appeared, these were transferred to Potato Rooting Medium PRM100c (Potato Rooting Medium containing only cefotaxime 100 mg/L; Appendix A). Subsequent multiplications were performed in Potato Propagation Medium (PPM; Appendix A) without antibiotics. When the plantlets were established, 3–5 leaves were cut for the corresponding DNA analysis.

### 4.5. Design and Selection of Guide RNAs (gRNA) for Potato Genome Editing

A dedicated tool to process genome information for the species that allowed for the generation of ‘gRNA pairs’ for efficient genome editing was built. The system was based on CRISPR-Analyzer [31] and CRISPETa [32]. The module CRISPR-Analyzer is a collection of command-line C++ scripts that enabled us to search and index all the possible “CRISPR sites” (i.e., protospacer + NGG sequences) in the *Solanum tuberosum* reference genome [41]. After individualizing each possible “CRISPR site”, CRISPR-Analyzer was used to compute the possible off-target sites with 0–4 mismatches, which is annotated as a mismatch pattern using the key (a,b,c,d,e). In this key, a-e indicate the number of off-target sites in the genome harboring 0–4 mismatches, respectively. CRISPETa is a suite of command-line Python scripts to find all possible gRNAs given a target genome region (in BED format). This tool also assigns a score to each gRNA according to its predicted on-target activity based on an empirical logistic regression model by Doench et al. [42]. Additionally, this module ranks and selects pairs of gRNAs according to the combined score of both guides and the specified maximum number of off-targets. This pipeline was adapted as a web application (available as Solanum tuberosum v4.03 in the “Genome Browser” option at www.fruit-tree-genomics.com, accessed on 20 March 2018) called the “Potato CRISPR Search Tool”, which, unlike these previous implementations, is an integrated system that includes the JBrowse Genome Browser [43] and SequenceServer [44]. This allows target sequences to be graphically selected in JBrowse using the “Highlight” tool in the menu bar; by using the specifically programmed plugin (“CRISPR” button in the menu bar), the target is loaded into the “Potato CRISPR Search Tool”. As in CRISPETa, the user can select a target region for a gRNA pair search while establishing advanced parameters such as the maximum permitted number of off-targets with 0, 1, 2, 3, and 4 mismatches and the individual and paired scores for the gRNAs. The tool we developed allows the resulting gRNA pairs to be examined for off-targets across the entire potato genome. The results for the off-target sequences are individualized according to chromosome, sequence coordinates, mismatch number and position, and location (exonic, intronic, or intergenic).

### 4.6. Gene Editing Vectors for Potato Genes

#### 4.6.1. Paired-gRNA Module Cloning into pGEF-U

Using the Potato CRISPR Search Tool, we chose those gRNA pairs located in distant exonic regions to further ensure loss of gene function, by deleting a large portion of the coding sequence (Appendix A). The selected gRNA pair (gRNA1 + gRNA2 for each gene) was incorporated into a PCR reaction to make the two-guided module using corresponding primers (DT1-BsF, DT1-F0 and DT2-BsR, DT-R0) for the edition of each gene according to the nomenclature described in Xing et al. [45]. The PCR reaction was performed using Platinum SuperFi proof-reading DNA polymerase (Thermo Fisher Scientific) according to the manufacturer’s specifications. The PCR mixture consisted of 2 µL DT1-BsF-Primer (20 μM), 2 µL DT1-F0-Primer (2 μM), 2 µL DT2-R0-Primer (2 μM), 2 µL DT2-BsR-Primer (20 μM), 10 µL 5× SuperFi Buffer, 1 µL dNTPs (10 mM), 1 µL pCBC-DT1DT2 (10 ng/µL), 0.5 µL Platinum SuperFi DNA Polymerase, and 29.5 µL nuclease-free water (final volume of 50 µL). PCR conditions were: initial denaturation 98 °C (30 s), 35 cycles of amplification (98 °C (10 s), 58 °C (10 s), 72 °C (30 s)), and final extension of 72 °C (5 min). Afterward, the PCR product (module) was electrophoresed and purified from ethidium bromide-stained 1.2% agarose gels with the ZymoClean Gel Recovery kit (Zymo Research) and eluted with 6 µL nuclease-free water. Then this module was cloned into pGEF-U by the Golden Gate reaction [46] through cloning modules using *Bsa*I restriction sites (Appendix A). The Golden Gate reaction mixture was as follows: 1 µL purified module (100 ng/ µL), 1 µL pGEF-U (100 ng/µL), 1.5 µL 10× T4 DNA Ligase Buffer (New England Biolabs; Ipswich, MA, USA), 1.5 µL 10× CutSmart Buffer (New England Biolabs), 1 µL *Bsa*I (New England Biolabs), 1 µL T4 DNA Ligase (New England Biolabs), and 8 µL nuclease-free water (final volume 15 µL). The reaction was performed according to the following incubation profile: 30 cycles of digestion/ligation at 37 °C for 10 min and 16 °C for 10 min, respectively, a final digestion at 55 °C for 25 min, and a denaturation round at 80 °C for 15 min. The ligated pGEF-X vectors (in which “X” represents either StPPO1, StPPO2, StvacINV1, or StBAM1 target genes, as appropriate; see Appendix A) were used to transform One Shot^®^ Top 10 *E. coli* chemically competent cells (Thermo Fisher Scientific); recombinant clones were selected on 100 mg/L kanamycin LB agar plates following the manufacturer’s instructions. *E. coli* clones carrying pGEF-X vectors (also referred to as Gene Editors) were checked by colony PCR, using Kapa Taq DNA Polymerase (Kapa Biosystems) according to the manufacturer’s specifications and using a primer pair named gmv-CI-Fw/gmv-CI-Rv (Appendix A) to confirm the two-guided module insertion into pGEF-U. Amplifications were resolved in 1.5% agarose gels followed by ethidium bromide staining. Positive clones were purified using the Zymo Miniprep Kit (Zymo Research) and sequenced (Macrogen Inc.) to verify a correct assembly, also using the gmv-CI-Fw/gmv-CI-Rv primer pair. pGEF-X clones were electroporated into electrocompetent *Agrobacterium tumefaciens* strain EHA105 as described above and used to transform leaf explants of ‘Yagana-INIA’, according to the previously described procedure [23].

#### 4.6.2. Validation of Guide RNAs

Editing pGEF-X vectors were validated in vivo using ‘Yagana-INIA’ leaf explants 7–10 d after gene transfer experiments. Three to five explants were chosen for DNA extraction, according to the method described in the Section 4.7 (“Genomic DNA Extraction”). These explants were those that showed the highest level of GFP fluorescence judged under epifluorescence microscopy. Isolated DNAs were subjected to PCR detection of edited versions of each gene using the check editing (CE) primers summarized in Appendix A (see also Appendix A). PCR assays were carried out using Kapa Taq DNA Polymerase (Kapa Biosystems) according to the manufacturer’s specifications. PCR mixes were as follows: 1 µL genomic DNA (50–100 ng), 2 µL 10× Buffer, 0.8 µL Primer Mix (10 µM), 0.4 µL dNTP (10 mM), 0.4 µL Mg^+2^, 0.08 µL Kapa taq, and 15.32 µL nuclease-free water, for a final volume of 20 µL. PCR conditions were: initial denaturation 95 °C (2 min), 40 cycles of amplification (95 °C (30 s), 58 °C (30 s), 72 °C (30 s)) and final extension of 72 °C (5 min). Amplifications were resolved in 1.5% agarose gels followed by ethidium bromide staining, and bands were purified from gels with the ZymoClean Gel Recovery kit (Zymo Research) and eluted with 6 µL elution buffer. PCR products were cloned into pGEMT-easy (Promega, Fitchburg, WI, USA) by ligation with the Ligase T4 enzyme (Promega) overnight at 4 °C; 3 µL of the ligation were used to transform *E. coli* One Shot^®^ Top10 chemo-competent (Thermo Fisher Scientific). The transformed cells were selected on LB medium and agar plates containing carbenicillin (100 mg/L), IPTG (0.128 mM), and X-Gal (32 mg/L). Positive clones were purified using the Zymo Miniprep kit (Zymo Research) and sequenced at Macrogen Inc. using the M13-Univ-Fw/M13-Univ-Rv primers (Appendix A).

### 4.7. Genomic DNA Extraction

‘Yagana-INIA’ genomic DNA was isolated from leaves using an adaptation of the protocol described by Steenkamp et al. [47]. Briefly, 100 mg samples were mixed with 700 µL CTAB extraction buffer and shaken for 3 min in a Mini-BeadBeater (Biospec Products Inc., Bartlesville, OK, USA). Extracts were incubated at 60 °C for 15 min, centrifuged for 10 min at 10,000 rpm, and kept at 4 °C. Six hundred microliters of the supernatant were transferred into a new sterile tube and gently mixed by inversion with 700 µL of a solution of chloroform:isoamyl alcohol (24:1 (*v*/*v*)). Samples were centrifuged for 10 min at 10,000 rpm at 4 °C and the aqueous phase recovered and gently mixed with 300 µL isopropanol. The mix was incubated at room temperature for 10 min. Samples were centrifuged for 15 min at 10,000 rpm at 4 °C. The DNA pellets were washed with 500 µL ethanol 70% (*v*/*v*) for 10 min, dried, and resuspended in 30 µL nuclease-free water, containing RNAse A at a final concentration of 100 μg/mL. DNA samples were incubated at 37 °C for 30 min and DNA was quantified and stored at −20 °C until use.

### 4.8. GFP Detection in Potato Explants

Observation and recording of the fluorescence emission by GFP were carried out using an epifluorescence microscope (Zeiss Axioscope Lab A.1 equipped with Filter Set 09, BP 450–490 nm and Filter Set 38, BP 470/540 nm; Zeiss, Oberkochen, Germany). The light source was provided by a 470 nm LED lamp. Images were acquired with a Canon Rebel T3 camera using the software EOS Utility (Canon Inc., Tokyo, Japan).

### 4.9. Verification of Edited Lines

Once candidate lines were well established as indicated in Section 4.4 (“Plant Regeneration”), 3–5 leaves from these plantlets were cut for analysis. The DNA extraction was as mentioned in the Section 4.7 (“Genomic DNA Extraction”). Isolated DNAs were used in PCR amplifications using CE primer pairs (see Appendix A) and the size of edited amplicons was evaluated. Reactions were carried out using the enzyme Phusion High-Fidelity DNA Polymerase (Thermo Fisher Scientific) according to the manufacturer’s specifications. The amplification mixture was as follows: 1 µL genomic DNA (25 ng), 4 µL 5× Phusion HF Buffer, 0.8 µL primer mix (10 µM), 0.4 µL dNTP (10 mM), 0.2 µL Phusion DNA Polymerase, and 13.6 µL nuclease-free water to a final 20 µL volume. The thermal profile was: initial denaturation 98 °C (30 s), 35 cycles of amplification (98 °C (10 s), 58 °C (10 s), 72 °C (10 s)) and final extension of 72 °C (5 min). Three microliters of this amplification reaction were cloned into pCR^TM^4Blunt-TOPO^®^ (Thermo Fisher Scientific) following the manufacturer’s instructions. Ligation mixtures were incubated for 5 min at room temperature. Three microliters of ligation mix were used to transform *E. coli* One Shot^®^ Top10 chemo-competent cells. Positive clones were sequenced using M13-Univ-Fw/M13-Univ-Rv primers (Macrogen Inc.).

## 5. Conclusions

We designed pGEF-U, a traceable LSL-based vector that leads to gene editing in potato through a primary selection of GFP emission from the explants subjected to a regular organogenesis regeneration procedure. Also, taking advantage of the multi-gRNA expression capacity in the vector, we aimed at the inactivation of genes associated with postharvest disorders in tubers. As proof of concept, both characteristics were used to define a straightforward method to identify edited individuals within the generated population. Sequencing studies of editing target points of both plants and calli encouraged further analyses of the plant population considering the particular action directed by each independent gRNA.

## Figures and Tables

**Figure 1 plants-10-01882-f001:**
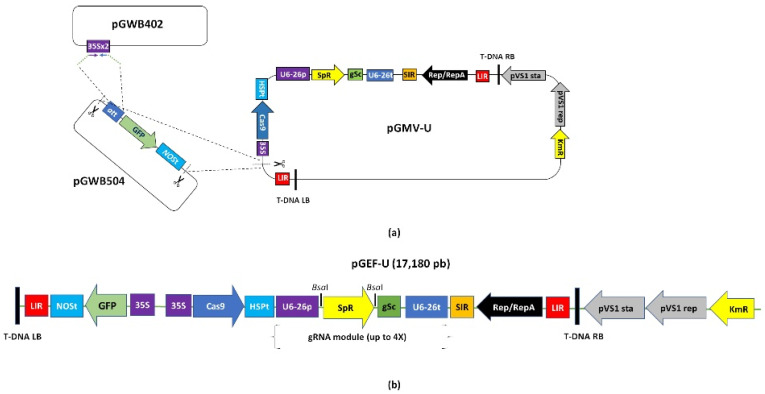
Universal Fluorescent Editor Geminivirus-based plasmid (pGEF-U). An LSL vector based on the *Bean yellow dwarf virus* (BeYDV) and harboring a *Green Fluorescent Protein* (*GFP*) expression cassette was built for gene transfer experiments in potato leaf explants. Based on its predecessor (pGMV-U [22]), a *GFP* expression cassette (from pGWB504) was inserted after proper recombination of a CaMV 35S promoter (from pGWB402) into the pGMV-U by traditional PCR-based cloning using the unique *Asc*I restriction enzyme site (scissors) (**a**). pGEF-U includes the sequence required for the insertion of up to four gRNAs (gRNA scaffold) (**b**). T-DNA RB, right border of the Agrobacterium T-DNA; LIR, large intergenic region from the *Bean yellow dwarf virus* (BeYDV); CaMV 35S, 35S promoter from *Cauliflower mosaic virus*; Cas9, Cas9 *Arabidopsis thaliana* codon usage; HSPt, terminator for *HEAT SHOCK PROTEIN 18.2* gene from *A. thaliana*; U6-26p, *A. thaliana* U6-26 RNA polIII promoter; SpR, spectinomycin resistance gene; KmR, kanamycin resistance gene; gSc, gRNA scaffold sequence; U6-26t, *A. thaliana* U6-26 RNA polIII terminator; SIR, short intergenic region from BeYDV; Rep/RepA, nucleotide sequence for the Rep/RepA replication genes; LB T-DNA, left border of the Agrobacterium T-DNA; *att*, recombination Gateway technology sites; *Bsa*I, restriction enzyme sites for additional gRNA expression cassettes.

**Figure 2 plants-10-01882-f002:**
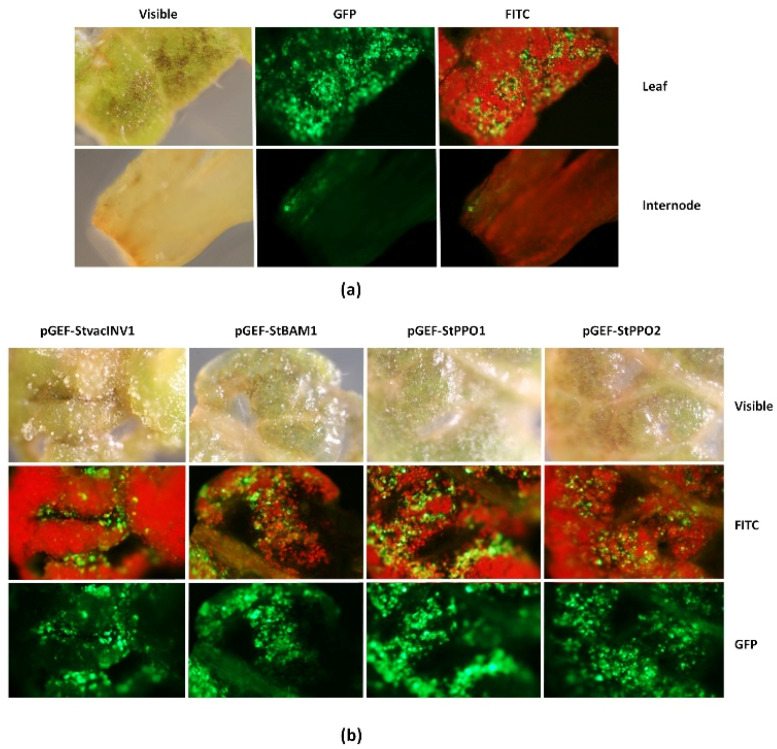
Functionality of pGEF-U in *Agrobacterium*-mediated transformation of ‘Yagana-INIA’ explants. Transient expression of pGEF-U was followed by GFP epifluorescence analyses at 7 days post-inoculation in leaf and internode sections of ‘Yagana-INIA’ (**a**). Once functionality was verified, dedicated vectors targeting specific target genes associated with cold induced sweetening (pGEF-StvacINV1 and pGEF-StBAM1) and polyphenoloxydase enzymatic browning (pGEF-StPPO1 and pGEF-StPPO2) were assembled and evaluated by leaf agroinfiltration and evaluated at this same dpi (**b**). Visible, field observation of the explants using white light; FITC, field observation applying the fluorescein-5-isothiocyanate filter under epifluorescence excitation; GFP, field observation applying the green fluorescent protein specific filter under epifluorescence excitation. Images were acquired under 50× amplification using an Axioscope Lab A.1 (Zeiss) system.

**Figure 3 plants-10-01882-f003:**
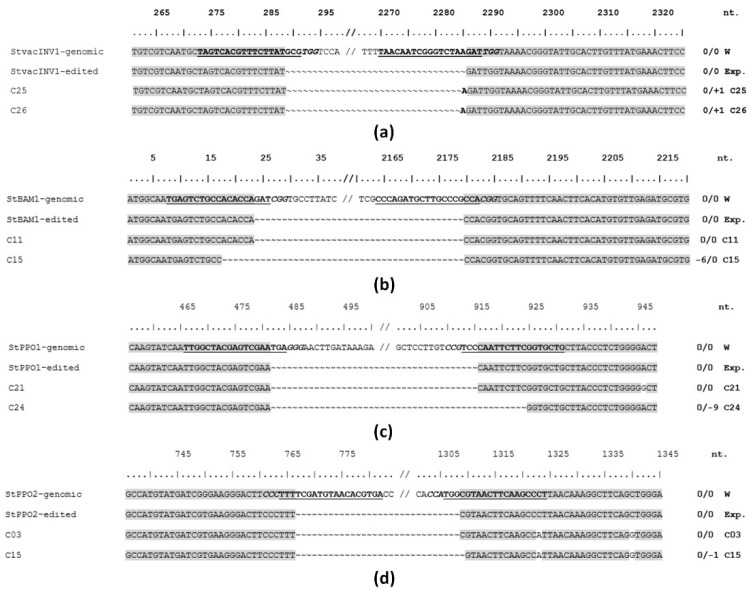
Targeted deletion gene editing characterization by the specific pGEF-X vectors. After infiltration, genomic DNA was extracted from 3–5 explants with the highest GFP and subjected to PCR using the characterization primers. The amplification products were resolved by gel electrophoresis, and the smaller bands, in comparison to each original gene version, were cloned and sequenced. Different DNA repairs at both ends after the CRISPR/Cas9 editing were obtained for the *S. tuberosum VACUOLAR INVERTASE 1* (*StvacINV1*) (**a**), *BETA-AMYLASE 1* (*StBAM1*) (**b**), *POLYPHENOL OXYDASE 1* (*StPPO1*) (**c**) and *2* (*StPPO2*) (**d**) genes. Wild type versions of the genes (genomic) are compared with their theoretically edited (Exp) and two experimentally detected (Cx) versions. Guide RNA sequences are in bold underlined font; protospacer adjacent motifs (PAM) are in bold italic font. Homologue nucleotides are boxed in grey. Nucleotide deletions are indicated by the “~” symbol; nucleotide insertions are in bold font. Total insertions/deletions in the amplified fragments are indicated for each Cx case.

**Figure 4 plants-10-01882-f004:**
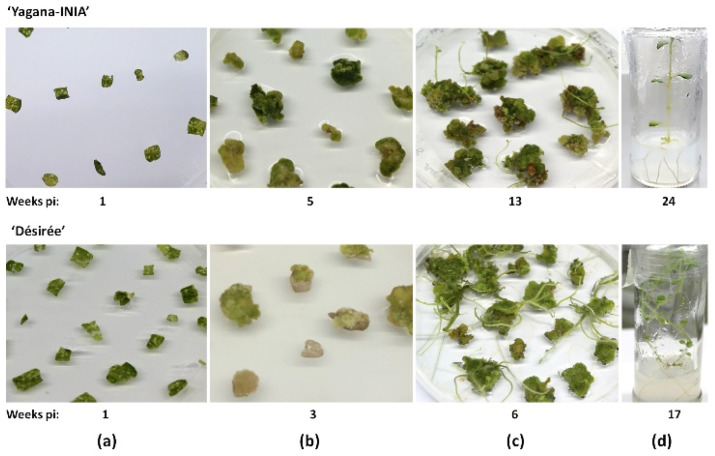
Regular procedure for ‘Yagana-INIA’ plantlet regeneration. Leaf explants are selected from in vitro plants aged 4–6 weeks old. Selected leaves are sliced into proper square-shaped explants. The first week, the explants are kept in the potato callus induction medium (PCM). (**a**) Then explants were transferred to potato shooting induction medium (PSM) to further induce callusing (**b**) and, subsequently, budding (**c**); finally, plantlet generation is completed when early shoots were transferred to potato rooting medium (PRM) (**d**). ‘Désirée’ is shown in the lower panel as a comparative procedure. Weeks pi, week post-Agrobacterium inoculation.

**Figure 5 plants-10-01882-f005:**
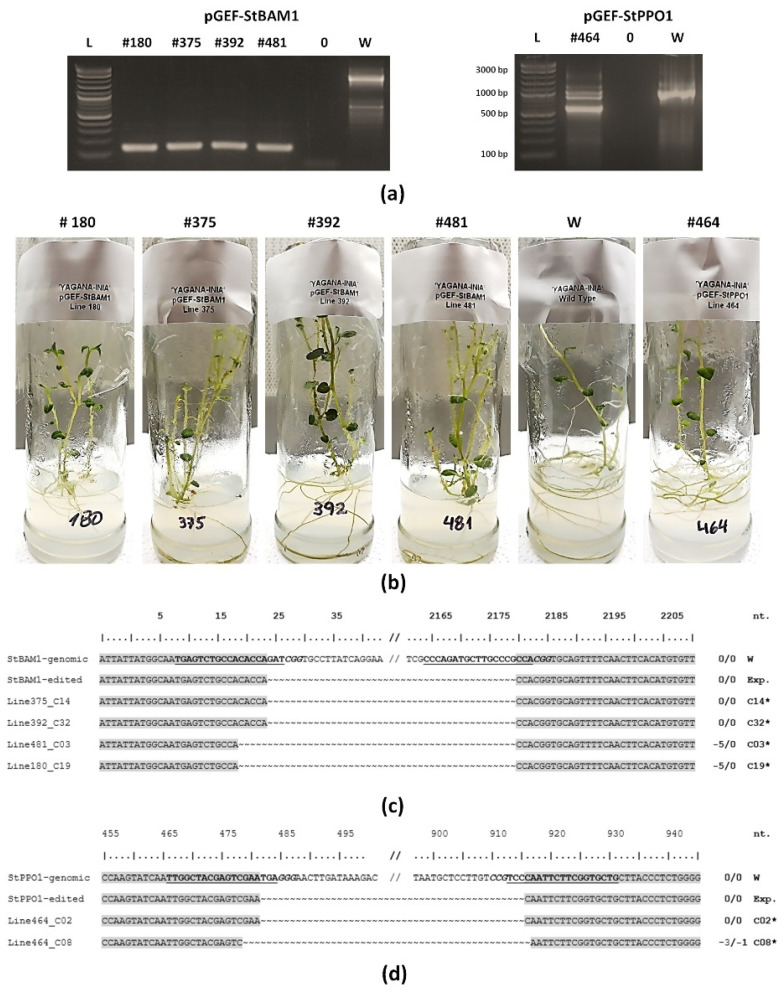
Targeted deletion gene editing in ‘Yagana-INIA’ plants produced by explants selected by transient GFP expression. Gene transfer experiments were performed in leaf explants using *Agrobacterium tumefaciens* EHA105 harboring the pGEF-X vectors (for *StBAM1* and *StPPO1* genes). Explants with higher GFP expression at 7 dpi were selected and led to regeneration. Plants from explants losing GFP expression in the process were individualized and subjected to genomic DNA extraction and PCR (**a**) analysis of the targeted genes. Selected double cut edited individuals (**b**) were subjected to further sequencing of the amplicons derived from PCR amplifications for *StBAM1* (**c**) and *StPPO1* (**d**) genes. Guide RNA sequences are in bold underlined font; protospacer adjacent motifs (PAM) are in bold italic font. Homologue nucleotides are boxed in grey. Nucleotide deletions are indicated by the “~” symbol. Cx, colony number of the sequenced bacterial clone resulting from PCR band cloning process. Total deletions in the amplified fragments are indicated for each Cx case. *, representative of two or more different sequenced colonies.

**Figure 6 plants-10-01882-f006:**
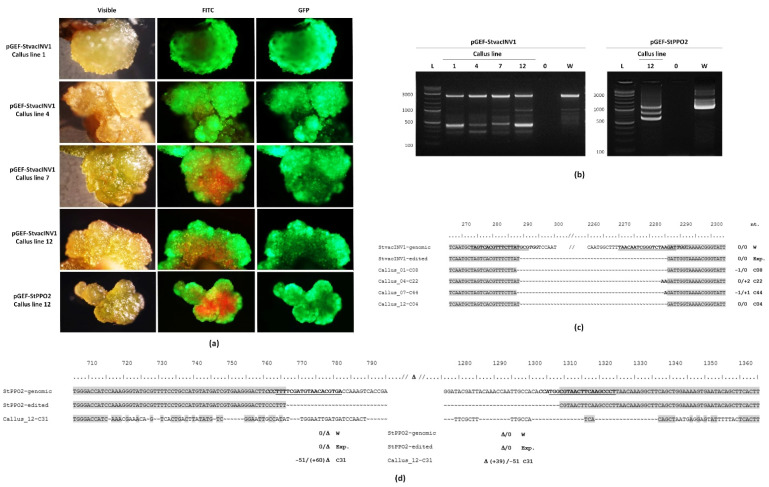
Gene editing in ‘Yagana-INIA’ calli produced by explants selected by stable GFP expression. Gene transfer experiments were performed in leaf explants using *Agrobacterium tumefaciens* EHA105 harboring the pGEF-X vectors (for *StvacINV1* and *StPPO2* genes). Explants with higher *GFP* expression at 7 days post-inoculation were selected and led to regeneration. Calli from explants keeping GFP expression during the complete process were individualized (**a**) and subjected to genomic DNA extraction and PCR for analysis of the targeted genes and transgene insertion of different vector components (**b**). Selected targeted deletion edited cell lines were subjected to further sequencing of the amplicons derived from PCR amplifications for *StvacINV1* (**c**) and *StPPO2* (**d**). Guide RNA sequences are in bold underlined font; protospacer adjacent motifs (PAM) are in bold italic font. Homologue nucleotides are boxed in grey. Nucleotide deletions are indicated by the “~” symbol; punctual nucleotide insertions are in bold fonts (**c**), large insertions are in regular fonts (**d**). Cx denotes the colony number of the sequenced bacterial clone resulting from the PCR band cloning process. Total insertions/deletions in the amplified fragments are indicated for each Cx case. Primers StvacINV1-CEa-Fw and StvacINV1-CEa-Rv were used for *StvacINV1* characterization in (**b**) (Appendix A).

**Table 1 plants-10-01882-t001:** Selected guide RNAs for potato genome edition.

gRNA Name	gRNA Sequence	gRNA Position	Size of the Expected Deletion
gRNA1-StvacINV1B	TAGTCACGTTTCTTATGCG	273–291	1997 bp
gRNA2-StvacINV1B	TAACAATCGGGTCTAAGAT	2270–2288	
gRNA1-StBAM1A	TGAGTCTGCCACACCAGAT	8–26	2156 bp
gRNA2-StBAM1A	CCCAGATGCTTGCCCGCCA	2164–2182	
gRNA1-StPPO1B	TTGGCTACGAGTCGAATGA	466–484	434 bp
gRNA2-StPPO1C	CAGCACCGAAGAATTGGGA	913–931	
gRNA1-StPPO2C	TCACGTGTTACATCGAAAA	764–782	542 bp
gRNA2-StPPO2C	AGGGCTTGAAGTTACGCCA	1306–1324	

**Table 2 plants-10-01882-t002:** Individualized ‘Yagana-INIA’ shoots selected from leaf explants subjected to gene editing by pGEF-X vectors.

pGEF-X X:	Initial Explants	GFP+ Leaves (%)	GFP+ Leaf Explants in PSM400cc Day 7 (pi) * (%)	Individualized Shoots	Shoots per Callus
StvacINV1	23	18 (78.3)	4 (17.4)	15	3.8
	35	16 (45.7)	10 (28.6)	61	6.1
	80	31 (38.8)	23 (28,8)	48	2.1
StBAM1	65	33 (50.8)	22 (33.8)	74	3.4
	101	54 (53.5)	26 (25.7)	52	2.0
	61	12 (19.7)	5 (8.2)	2	0.4
StPPO1	58	45 (77,6)	39 (67.2)	86	2.2
	200	126 (63.0)	84 (42.0)	52	0.6
	81	57 (70.4)	36 (44.4)	3	0.1
StPPO2	26	18 (69.2)	14 (53.8)	37	2.6
	263	152 (57.8)	79 (30.0)	122	1.5
	47	18 (38.3)	15 (31.9)	11	0.7

* Positive selection through GFP expression monitored at 7 dpi.

**Table 3 plants-10-01882-t003:** Yagana-INIA edited transgenic calli obtained by *GFP* expression.

Gene Editor	Initial Explants	GFP Positive	PSM400cc Day 7 (pi)	PSM200cc Day 82 (pi)	PSM200cc Day 271 (pi)	Edited TG Calli *
pGEF-StvacINV1	93	51	32	12	6	4
pGEF-StPPO2	62	29	19	14	3	1

* Confirmed by PCR and sequencing.

## Data Availability

The Potato CRISPR Search Tool is available at www.fruit-tree.genomics.com, accessed on 20 July 2020, menu Biotools.

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
