# Peer review of "A Traceable DNA-Replicon Derived Vector to Speed Up Gene Editing in Potato: Interrupting Genes Related to Undesirable Postharvest Tuber Traits as an Example"

_plants, 2021, doi:10.3390/plants10091882_

Round 1
Reviewer 1 Report
General comments
The manuscript presents a courageous approach to genetic manipulation in plants, more precisely in potato, by using a traceable vector derived from DNA-replica, in order to break the genes involved in the formation of negative traits that occur in the process of tuber preservation.
The design and evaluation of the LSL-based vector T-DNA pGEF-U, designed to allow genome editing when a traceability manufacturer, Green Fluorescent Protein 109 (GFP), allows monitoring of vector-cell interaction during the process, is presented through the paper. For the evaluation of the system, genes associated with unwanted post-cancer events that cause enzymatic and non-enzymatic browning of potato tubers were targeted.
Specific issues
We have identified a few things that can be improved.
- The title of the paper can be presented in a simple, less detailed wording.
- It is usually not recommended to use the abbreviations in the abstract.
- The introduction contains few bibliographic references to the current state of knowledge in the field, I think it should be updated and brought to the level of 2021.
For the rest things are fine, correct research methods, data processing and their interpretation is appropriate to the scientific level of the publication, the comparative analysis with other results is well carried out.
The graphical presentation of the results is especially interesting.
By improving the introduction part of the manuscript, it will also expand and improve the references.
Congratulations and good luck!
Reviewer 2 Report
Please see attached
